# Psychoanalysis of COVID-19 Patient Narratives: A Descriptive Study

**DOI:** 10.3390/medicina59040712

**Published:** 2023-04-05

**Authors:** Yu Deng, Luxue Xie, Li Wang, Yaokai Chen

**Affiliations:** 1College of Language Intelligence, Sichuan International Studies University, Chongqing 400031, China; dengyu@sisu.edu.cn; 2School of English Studies, Sichuan International Studies University, Chongqing 400031, China; 3Science and Education Department, Chongqing Public Health Medical Center, Chongqing 400036, China; 4Department of Infection Diseases, Chongqing Public Health Medical Center, Chongqing 400036, China

**Keywords:** COVID-19 patients, Lacanian desire theory, psychoanalysis, narratives, mental health

## Abstract

*Background and Objectives*: COVID-19 patients are a psychologically vulnerable patient group who suffer from both physical symptoms and psychological problems. The present study is a psychoanalytic investigation of COVID-19 patients utilizing Lacan’s desire theory. We aimed to explore the manner in which patients’ desire is presented in their lived experience narratives and sought to discover factors which directly impacted on this process. *Materials and Methods*: In-depth semi-structural interviews were conducted with 36 COVID-19 patients in China. During each interview, participants narrated their lived experiences of COVID-19 infection. Emotions, metaphors, and behaviors in patient narratives were collated as the main points for psychoanalysis. *Results:* Our findings demonstrated that the desire for being a healthy person made patients emotionally sensitive to the social environment. Anxiety and obsessive behaviors emerged in the process, which reveals their desire for that which they lack. Furthermore, public fear with respect to COVID-19 was somehow converted to psychological pressure on COVID-19 patients. Thus, these patients attempted to “de-identify” their identity as “patients”. Positive responses of COVID-19 patients to the external world included admiring medical personnel, government, and country, while negative responses included interpersonal conflicts or complaints about discrimination. Following the rules of the Other, COVID-19 patients were influenced by the Other’s desire in constructing their own image of a healthy person. *Conclusions*: This study revealed COVID-19 patients’ psychological need to rid themselves of the identity of “patient” at the individual and social level. Our findings have clinical implications in helping COVID-19 patients to reshape their identity and to live a normal life.

## 1. Introduction

Subsequent to the outbreak of COVID-19, a growing body of literature has explored the psychological problems of COVID-19 patients [1,2,3,4,5]. The most common post-COVID mental health problems among COVID-19 patients include sleep disorders, anxiety, depressive symptoms, fatigue, memory problems, and difficulty concentrating. Critically ill patients account for the largest number of psychiatric consultations [6]. Parikh et al. evaluated the mental health of COVID-19 patients in India and results showed that symptomatic patients were relatively more psychological distressed than asymptomatic patients [7]. Alamri et al. investigated the mental health of COVID-19 patients in Saudi Arabia and observed that 50% of COVID-19 patients were considered to have anxiety and depression or demonstrated a borderline level of anxiety and depression [8]. Kang et al. assessed 107 COVID-19 patients treated in a community treatment center in Korea [9]. All patients showed mental health problems including depression, anxiety, post-traumatic stress disorder (PTSD), and somatic symptoms. Gu et al. evaluated the psychological status among COVID-19 patients treated in Fangcang shelter hospitals in China, and their results indicated that COVID-19 patients suffer from severe symptoms of mental distress such as anxiety, depression, insomnia, and perceived stress [10].

Most recently, lived experience narratives have been found to be an effective marker to detect COVID-19 patients’ mental health status [11,12,13,14,15]. Sahoo et al. explored the psychological distress encountered in COVID-19 patient narratives and found that infected patients had symptoms such as anxiety, fear, worries, and abnormal behaviors [11,15]. Drawing on the narrative analysis of the lived experience of COVID-19 patients in Denmark, Missel et al. observed that patients had a fear of being a burden to others and of infecting others [12]. They talked of the illness in terms of infection and relationships, including “COVID-19 as a threat to existence”, “COVID-19 as a threat to bodily perception”, and “COVID-19 as an interference with social relationships”. Deng et al. investigated narratives of COVID-19 patient who retested positive after recovery in China [13]. Narrative analysis showed that patients felt anxious, guilty, nervous, fearful, and lost. Their re-positivity worsens such negative emotions and makes them more stressful than others.

Psychoanalytic studies during the COVID-19 pandemic have inspired research related to the mental health of COVID-19 patients. Velykodna observed that countertransference is related to real losses during the COVID-19 pandemic and psychoanalysts should pay adequate heed to the phenomenon of countertransference [16]. Haber pointed out that the stresses of COVID-19 evoke a return to archaic overburdening and threatened loss of selfhood [17]. Patients are shaken by the loss of relationships in isolation. Gairola reviewed the impact of the COVID-19 pandemic on psychosis and observed that patients who are suffering from severe mental illness show more anxiety than healthy controls [18]. Hence, stressful events such as the COVID-19 pandemic can trigger psychotic disorder relapse.

To date, although extensive surveys and narrative research has been conducted to address mental health issues during COVID-19, there has been limited investigation of psychoanalysis in COVID-19 patients. The published literature regarding psychoanalysis and COVID-19 concentrated on the general public and these studies are confined to reflections and theoretical intuition, while empirical psychoanalysis of COVID-19 patient narratives is underestimated. COVID-19 patients are explicitly affected by the infection. They demonstrate psychological problems such as anxiety and depression during the quarantine period. Psychoanalysis provides a new pathway to explore COVID-19 patient mental states through their emotions, metaphors, and behaviors as seen in the lived experience narratives. This study is an attempt to utilize psychoanalysis in COVID-19 patient narratives on the basis of Lacan’s desire theory. It is hypothesized that COVID-19 patients display abnormal behaviors and mental health symptoms due to the desire for lack (i.e., desire for a fulfilling of the lack, or desire for that which they lack). In desire theory, patients are affected by the representation of the Other’s desire, namely, being healthy and clean. The desire for lack makes patients respond as what in their narratives. Patients demonstrate strong desire to be admitted and be identified as a healthy person. The present study aimed to explore their emotional needs and explain their psychological status through psychoanalysis of patient narratives and sought to discover factors which directly impacted on this process. Specifically, we address three questions: (1) What are the internal factors and external factors of the mental states of COVID-19 patients? (2) How do these factors work to reflect patients’ psychological requirements? (3) What are the patients doing to reconstruct their identity?

Regarding the first question, the internal factors pertain to the ego, or a question about “who am I”? The internal state may point to negative emotions such as patients’ fear of COVID-19. External factors relate to the environment where others’ fear of COVID-19 is converted into discrimination. The second question considers patients’ psychological requirements in terms of labeling and de-labeling. Positive and negative emotions are manifested in the process of labeling and de-labeling. The third question concerns patients’ desire of rebuilding their identity. COVID-19 patients are trying to be recognized by others and be identified as a healthy person in mind.

This study employed semi-structured telephone interviews with 36 individuals infected with COVID-19 in China by convenience sampling from September 2020 to March 2021 to collect lived experience narratives. The phenomenological qualitative interview method can offer insights into patients’ subjective syndromes [13,14]. Emotions, metaphors, and behaviors in interview data were coded for psychoanalysis in line with Lacan’s desire theory in order to reveal patients’ psychological problems.

## 2. Lacanian Desire Theory

Lacan distinguishes desire from need and demand. Need pertains to biological instinct, while demand articulates need and acts as the demand for love [19]. Desire is produced by the articulation of need in demand. Lacan holds that “Desire is neither the appetite for satisfaction, nor the demand for love, but is the difference that results from the subtraction of the first from the second” [19]. This implies that desire begins to take shape in the margin in which demand becomes separated from need. Man’s desire is endowed with meaning by the Other’s desire. A need can be satisfied but desire cannot. Desire can never be satisfied and is like an endless journey where one cannot see the goal. Desire is concerned with lack, which is beyond explicit expression. It is only represented as a reflection on a veil. Desire is a function central to all human experience [20]. According to Lacan, the desire of lack can never be satisfied. “Being comes into existence as an exact function of this lack. Being attains a sense of self in relation to being as a function of this lack, in the experience of desire”. The absence of recognition results in a broken self of the subject. In Lacan’s argument, man’s desire entails: (1) desire as others’ desire; (2) desire for others’ desired object; (3) desire as the object of others’ desire.

### 2.1. Desire as the Other’s Desire

Desire is the Other’s desire. It is the desire for recognition. “For this desire itself to be satisfied in man requires that it be recognized, through the accord of speech or the struggle for prestige, in the symbol or the imaginary” [19]. For example, COVID-19 patients are very likely to complain of discrimination from others. They tend to take some actions to avoid such a situation or to make some changes for recognition. Patients require people to change their minds through specific behaviors, as patients desire to be recognized by others.

### 2.2. Desire for the Other’s Desired Object

The desire is about attaining the object of the Other’s desire. The object is what is desired by others. Lacan mentions that “the fact that the subject’s unconscious is the Other’s discourse appears more clearly than anywhere else in the studies of Freud” [19]. However, the object of his/her desire is a point of the desire. It is important to identify the subject with whom he/she identifies. It is the desire for others’ desired object. For instance, the desire of COVID-19 patients consists of seeking the image of a healthy person. However, the criteria as to whether the patients are cured or not may largely depend on others’ evaluation.

### 2.3. Desire as the Object of the Other’s Desire

The desire is the object of the Other’s desire. “Man’s desire finds its meaning in the Other’s desire, not so much because the Other holds the keys to the desired object, as because his first objective is to be recognized by the Other” [19]. The Other’s recognition is the embodiment of the Other’s desire. In the present study, the Other’s recognition can be labeled with an image of a “healthy” or “harmless” person. Thus, COVID-19 patients try to form such an image to be recognized. Patients’ abnormal behaviors demonstrate their efforts to fulfill their desire to recognize themselves and to be recognized by others.

## 3. Material and Methods

### 3.1. Research Design

This study employed qualitative, phenomenological methodology to explore the lived experience of COVID-19 patients. Specifically, in-depth semi-structural interviews were conducted with persons infected with COVID-19. Emotions, metaphors, and behaviors in patient narratives were taken as the main points for psychoanalysis in terms of Lacanian desire theory. The psychoanalysis of patient narratives can reveal subjectivity syndrome of COVID-19 infected persons.

### 3.2. Participants

The participants in the present study were 36 Chinese COVID-19 patients who had been in isolation at Chongqing Public Health Medical Center for treatment from September 2020 to March 2021. They were recruited through convenience sampling. The inclusion criteria were a confirmed diagnosis of COVID-19 and the presence of the symptoms of COVID-19 disease. Furthermore, all patients were admitted to the hospital for quarantine and treatment [13]. Nine participants were still hospitalized when they attended the interview. Six patients had retest-positive experiences after their recovery. These patients had different social backgrounds and genders (see Table 1). There were 16 females and 20 males. The mean age was 38. Among the 36 patients, the longest hospital quarantine time was 69 days (sub 14, male), while the shortest was 2 days (sub 30, female, she was still in isolation when she attended the interview). The mean quarantine time is 16 days.

Written informed consent was obtained from all participants before the interview. The study was approved by the Ethics Committee of Chongqing Public Health Medical Center (Approval Date: 4 September 2020; Number: 2020-048-02-KY)

### 3.3. Procedure of Interviews

Semi-structured interviews were conducted with the 36 COVID-19 patients. The procedure is shown below in Figure 1. The interviews were conducted by two interviewers from September 2020 to March 2021. Patients signed informed consent at the hospital to learn the purpose of the study. All patients had the right to freely decide whether to participate in the interview or not. We then accessed the medical information of patients from the hospital database. The next step was to contact the patients. They were free to ask questions about the study. If the patients refused to continue with the interview, they had the freedom to withdraw at any time. Prior to conducting the interview, patients were informed that interview data were confidential and would be used for scientific research anonymously. Then, interviews were carried out with individual patients. Participants were asked questions concerning their lived experience of COVID-19 infection (see Appendix A). During the interview, patients answered the questions without interruption. They could narrate their experiences of COVID-19 in their own manner. Narration of the patients was audio-recorded. Finally, we obtained 1461.68 min of interview recording, with the mean length of 40.6 min for each interview. These patient data were stored in the hospital and can be accessed as a narrative corpus for scientific research and for designing clinical intervention schemes.

### 3.4. Data Analysis

The data were transcribed into text verbatim. Interviewer’s scripts were excluded, and patient narratives were reserved. Next, three coders identified the metaphorical narratives [21,22,23], the five types of emotional narratives (Happy, Sad, Fear, Angry, Disgust) [24,25], and the narratives of behaviors [22,23,24,25]. These narratives overlapped to some degrees. These narratives can demonstrate subjectivity syndrome of patients due to COVID-19 infection. For instance:


*“I felt that I was in darkness when I saw the COVID-19 pandemic. I don’t know how to say it. I think that our society and national policy are good. The virus is horrible”*

*(Patient 12, darkness and light metaphor)*



*“I felt my mind was blank. I could hear nothing whatever they said. …I was in panic when I could hear”.*

*(Patient 4, emotional narratives of Fear)*



*“I felt that I liked to take medicines. I thought I could get recovery only if I kept taking medicines. The only thing I liked was to take medicines. So I tried my best to get medicines. I must take medicines. If the doctor did not give me medicines, I worried that there was no hope, but I thought there was a hope if he gave me medicines“*

*(behavior narratives)*


Patient narratives were regarded as biomarkers for psychoanalysis. Next, three aspects were taken for psychoanalysis: (1) the relationship between COVID-19 patients’ lived-experience narratives and the desire; (2) the effects of external factors on patients’ mental health and their responses for desire; (3) the efforts of self-construction due to the sense of being broken.

The first aspect concerned patients’ desire for “clean”. COVID-19 patients repeatedly informed others that they were harmless after recovery. However, their infection experience had labeled them “unclean”. As a result, they were in a sense of broken, which was originated from others. The sense of broken turned into lack. Patients desired to be the object of the Other, namely, to be “clean” or a “healthy person”. Ultimately, the battle between recognizing and being recognized constituted the internal factor for patients’ psychological problems.

The second aspect was about how external factors impact patients’ internal mental state. For example, a patient (patient 10) said he was numb, sitting on his bed and thinking of nothing. He felt nothing even when a nurse drew his blood for examination. His response was the representation of sense of broken. This patient was discriminated against by people around him due to the contagious nature of the virus. To some extent, those people were executers of rules regarding the standard of a healthy human. The external pressure expanded the hole in the patient’s mind. Helplessness forced the patient to recollect the broken ego. Thus, external factors of patients’ psychological problems were somehow due to social relationships and external pressure.

The third aspect concerned the responses of COVID-19 patients to the internal and external effect regarding their psychological needs. For instance, one patient (patient 6) described that the infection made him a “monster”. The unhealthy state caused his self-broken and led to anxiety. The negative conditions persisted because of the desire of lack. His psychological needs of self-construction were largely affected by the internal and external factors.

## 4. Results

### 4.1. Quantitative Distribution of Patient Narratives

Metaphorical narratives were classified into different metaphor categories [21,22,23]. Table 2 presents the 308 metaphorical narratives concerning the psychological state of COVID-19 patients, including 48 metaphor categories. The top ten metaphors used by patients to narrate their psychological processes are image metaphor, motion metaphor, container metaphor, family metaphor, life and death metaphor, war metaphor, animacy metaphor, symbolic metaphorical enactments, metaphor of integrative behavior, and metaphor of darkness and light.

Five types of emotional narratives, namely, Happy, Sad, Fear, Angry, and Disgust, are shown in Figure 2. Generally, the negative emotional narratives versus positive emotional narratives comprise 64% versus 36% of responses, indicating that COVID-19 patients narrated significantly more negative emotional complex experience relating to infection. Notably, fear narratives accounts for the highest ratio among the negative emotions, suggesting that the virus infection has devastated COVID-19 patients’ psychological states.

Given that behavioral narratives were present in metaphorical narratives and emotional narratives, we do not demonstrate them separately here.

### 4.2. Qualitative Description of Patient Narratives

#### 4.2.1. The Relationship between Lived-Experience Narratives and the Desire

In Lacan’s concept, the ego is the illusion of a false wholeness based on taking oneself for another or an image in the mirror that forms the substratum for the ego in the “mirror stage” of development rather than a subject or seat of reality orientation [19,20]. In the symbolic dimension, the object needs to project himself/herself to the image in the mirror so that they can identify “me” with the help of the Other. However, the object leaves the mother’s body and means an eternal lack. The object constantly demands for a fulfilling of the lack. However, what the object demands is unable to be attained. Therefore, the ego, which is constructed by the image of the Other in the mirror, leaks out from the broken wholeness. Desire then emerges from the anxiety related to lack (See Figure 3).

According to Lacan, man’s desire is the desire of the Other. The desire is reformed by the object to identify the ego. He/she needs to respond to the Other’s desire so that the object can find their own desire [19,20]. Desire is embodied as the battle between recognizing and being recognized, and is implicated in COVID-19 patient narratives, including metaphors, emotions, and behaviors.

Metaphor plays a significant role in psychoanalysis. Since Freud, metaphor has expanded its foothold in psychoanalysis. As metaphor is ubiquitous, psychoanalysis in metaphor concerns how metaphors are chosen and what the relation is between symbol and fact. Generally, metaphor is “the linguistic bridge from body to mind” [26]. In this study, COVID-19 patients extensively used image metaphors to illustrate their image and desire so that they could not only be heard, but also be seen. For instance:


*“…during quarantine, the most thing I thought about was that I was serving a life sentence in prison. I had no idea when I could return home. Then, I got despaired. Everything in my mind was all about something like this.”*

*(Patient 7, image metaphor and trials, law metaphor)*



*“Later, I told him about that. During the next day, the door wasn’t locked on. I felt different when the door wasn’t locked. I didn’t know. If the door was locked, I felt like I was in jail. Just felt like a jail. If the door was only closed but not locked, I felt…Er…My mood would be a little different. They locked the door, and that really made me scared, because you didn’t know how long you would be isolated”*

*(Patient 10, image metaphor)*



*“When we were diagnosed with COVID-19, other people looked at us strangely…then they kept a long distance away us when we went for a COVID-19 test. They poked their head through the wall to check whether we were following up or not. …But I thought… I felt like a… a real… I noticed that, I mean, they were afraid of something like the plague. And I was the plague. The first time I had a feeling of being…the plague”*

*(Patient 20, image metaphor)*



*“We are victims rather than prisoners. Some people looked at me strangely. Because I am a forthright man, I said that we are not criminals. We are the victims of the pandemic”*

*(Patient 24, image metaphor)*


The COVID-19 pandemic sets a macroscopic scene in which the virus is evil, patients are dangerous, and health is recognized by society. People in the environment use language to prescribe “me”. However, COVID-19 patients are like a variant marked by the Other. The anxiety of lack emerges in patients’ minds and can be detected in their lived-experience narratives. The four patients described earlier all labeled themselves as a variant. They moved out of the normal trajectory as a healthy person. Other’s discrimination worsened their anxiety of lack, which pushed them to see the “black hole” of lack. The fact that people identify COVID-19 patients as unrecognized beings could likely break the shell of patients’ egos. When the internal needs are leaking out, COVID-19 patients demand satisfaction. In other words, they try to chase the desire of being a healthy person, which can fulfill the whole in reconstructing their broken ego.

In the narratives, specific behaviors illustrate the psychological symptoms of COVID-19 patients, such as anxiety and obsessive propensity. Behaviors are signs. “A Sign is anything which is determined by something else, called Object, and so determines an effect upon a person, called its Interpretant, that the latter is thereby mediately determined by the former” [27]. Behaviors depicted in patient narratives reveal patients’ responses to the internal and external world. For example:


*“Because of my physical constitution, all I wanted was to sleep. I did not remember something which made me unhappy. If you say that, I was worried about my family. I contacted with my children at home. I was very, very, very worried about this, and I felt scary during the 14 days. So, I constantly asked about their physical states every day since I feared that others would be infected by me.”*

*(Patient 2, specific behaviors, FEAR)*



*“…sometimes I could not fall asleep. If I failed to fall asleep, or I could not sleep at home, I feared that my brain would shut down when I was thinking too much. Then I always got up and ran around at home in the middle night, I wanted to end myself at that time. But I definitely could not do that because I have a child.”*

*(Patient 7, obsessive behaviors, FEAR)*



*“I could not sleep when I was waiting for the result of my COVID-19 test. I just wanted to get the report as soon as possible. I was so worried that I could not fall asleep. I worried about my family and my children, and myself. I think that maybe I was lucky and not infected at that time. It was just anxiety. Very anxious. I was anxious, and anxiety made me suffer from insomnia. Later, two hours later, I called them, asking about my result. Then I kept asking the same question for every two hours”*

*(Patient 8, obsessive behaviors, FEAR)*



*“I never thought that my result was negative. Then the doctor told me it was negative. (laughing) I was tested negative. So I asked the doctor when I could leave the hospital. He told me how I could ask this question since I received treatment in the hospital for just a few days. He said that I must take COVID-19 test twice or more if I wanted to leave hospital. Then I fell on the bed and told them to exam. 24 h later I took the COVID-19 test again. The result was negative again. Then I asked the doctor when I could leave the hospital. I said that I heard from others that I could leave hospital only with one negative result of COVID-19 test. But some people were in the hospital and some people said I needed to take the COVID-19 test twice. It was just…I must go out because I fear the COVID-19 test. Then I frequently asked the doctor: ‘Can I go out?’.”*

*(Patient 13, specific behaviors, FEAR)*



*“…We feared of the risk of infecting others. It seemed that our body got ill due to the COVID-19 infection, which made us fear of contact with others. If we wouldn’t infect others, everything was fine. But if we made others infected, we would feel guilty about this. We dared not contact others outside, or we would greet our neighbors in a long distance. We were afraid of contacting people. No contact. We feared of transmitting”*

*(Patient 26, specific behaviors, FEAR)*


The purpose of patients’ symbolic behaviors is to attain the perfect ego. Most COVID-19 patients in the present study demonstrated repetitive behaviors in their lived-experience narratives. Patients tended to transfer their anxiety into a specific behavior. In the preceding examples, patient 2 kept asking about his family’s physical state; patient 7 suffered from insomnia and ran in the middle night, showing a suicidal tendency. Patient 8 asked for her COVID-19 test result frequently due to excessive anxiety. Patient 13 continually asked the doctor about her date of hospital discharge. Patient 26 intentionally kept away from other people because she worried about transmitting the virus. The reason for their repetitive behaviors concerns recognition. Due to the uncertainties related to COVID-19, infected patients are very likely to be labeled as different beings in the social system. The Other prescribes a rule and “orders” patients to follow the rule; patients cannot identify themselves as “me” because they construct the ego from the Other’s desire. When patients’ desire for recognition (“a healthy person” in reality) could not be fulfilled in the external world, they were likely to seal “I” (the identity of patient) up in the internal world. Their anxiety of lack may then turn into a series of repetitive behaviors in the outside world, which reflected patients’ internal desire for a fulfilling of the lack (i.e., “perfect ego”).

During COVID-19, society is like a mirror. Everyone needs to be recognized by the Other. Man’s desire is endowed meaning by Other’s desire. In the situation of COVID-19 infection, anxiety of lack results in a strong sense of stigma in COVID-19 patients’ mind. The shame of identity (i.e., COVID-19 patients) worsens patients’ psychological state. For instance:


*“Some psychologists called me when I was discharged from the hospital because I couldn’t fall asleep at that time. I felt that the virus was everywhere”*

*(Patient 2, presence, accompanying, and absence)*



*“I felt that the virus was everywhere. That was my feeling at that time.”*

*(Patient 6, presence, accompanying, and absence)*


The two metaphors of presence and accompanying clearly reveal patients’ negative emotions concerning the sense of shame of their identity as COVID-19 patients. Such shame and anxiety come from their sense of insecurity because COVID-19 patients contract the illness accidently. The accidental infection creates pain both physically and psychologically in patients. They are unable to escape from COVID-19 although they seek to tear off the virus label attached to them. A completely healthy illusion calls them to change and construct a new ego to fit the image of the Other. Patients’ shame and anxiety result from the self-consciousness of needing the Other, of a deep dependency on the relationship [28]. Then, the desire of the Other (i.e., ridding themselves of the identity of patients) emerges in patients’ mind. The alienation of being a healthy person constantly urges COVID-19 patients to complete the purpose in meeting the desire of the Other.

#### 4.2.2. The Effects of External Factors on Patients’ Mental Health and their Responses for Desire

Humans are social creatures. We construct the image of ourselves by connecting to the external world. COVID-19 patients desire change and show a strong desire to be recognized by others. The signals from others illuminate an image of recognition to drive the object’s internal desire. COVID-19 patients place themselves in an inferior position partly due to negative influence from the external environment. For instance:


*“They would complain. Some people complained. But they didn’t think that you were a bane of others, or you gave them troubles deliberately. However, they wouldn’t look at the thing with an objective perspective”*

*(Patient 9, behavior of others)*



*“…we walked on the way. We took a walk at night. We walked on the same road…he came to us. Then he saw us when we were walking on the left side. Thump, this man rushed to the right side. Just like this. Then as he saw me, he covered his mouth immediately, and his nose. He ran away”*

*(Patient 18, behavior of others)*



*“Those residents quarreled with us. One of them said that he would beat me to death if we walked in (the community). I was angry then. For those healthy people, they rejected us extremely.”*

*(Patient 27, behavior of others)*



*“…of course, I was in a bad mood. Many local people bought houses in Yun Ji town. Although some people knew us, they dared not greet us. They just took us as strangers. No greetings. Just, scary.”*

*(Patient 27, behavior of others)*


In the preceding extracts, people around COVID-19 patients showed fear of contagion. Keeping a distance was a way to avoid infection. What made these people take extreme defensive behaviors resides in their negative image of COVID-19. When they saw COVID-19 patients, they retrieved the negative information about the virus. The negative information was imprinted in their minds and provoked them to form an abnormal behavioral pattern. In doing so, they protected themselves and they maintained an identity as a healthy person through specific actions such as complaining, quarreling, or intentionally keeping a distance. This worsened the sense of broken among COVID-19 patients. When people created the health illusion, it showed the Other’s desire. In this sense, COVID-19 patients repeatedly de-stigmatized to fulfill the lack of desire because the Other’s desire was their desire. In the mirror of society (i.e., the macro-system), COVID-19 patients wish to reconstruct the broken ego with the help of the Other’s image, namely, a healthy being. Lacan deems that man’s desire finds its meaning in the Other’s desire, not so much because the Other holds the keys to the desired object, but because his first object(ive) is to be recognized by the Other [19]. Because of the contradiction between recognition and being recognized, COVID-19 patients were sometimes relatively aggressive in their quest to reject their label and proved their “innocence”. For instance:


*“I quarreled with the disease control professionals. Their work is to take a blood sample. Why did I quarrel with them? Because it was not long that I left the hospital, and I lived in hotel for a day after de-isolation. But they brought me to have a COVID-19 test three days later. I said to them that I was not tested positive. What did you think? I said that I just felt better for discharging from hospital. I just felt better when I lived in hotel. You guys took a sample every six or seven days. Was it anything different about my sample? Why did you take the sample every day? This made me upset. Didn’t you have other things to do? You were wasting the government finance. Did you think that there was too much money? They said that they were unwilling to do this. I didn’t think so. I complained: “why did not you stand in my shoes to think about what I thought. I quarreled with them that day. But I had no choice, and COVID-19 test was necessary. So, I still supported their work”*

*(Patient 16, specific behavior, ANGRY/FEAR)*


In this case, patient 16 hated the process of taking blood samples frequently. He thought that the professionals still treated him as a COVID-19 patient. His quarreling referred to the hatred of being labeled as a patient. His desire for the Other pushed him to fit the image of a healthy person. Thus, he felt angry when he was considered to be a patient after hospital discharge.


*“…I felt upset. I would not take phone calls or video call on wechat. My friends and family sent me messages: “we hear that you are retested positive.” “how does it happen?” Nobody had the patience to respond. In the hospital, I felt…I was a person who had a good attitude. But then, I…uh, I wanted to curse.”*

*(Patient 18, specific behavior, SAD)*


The external environment somehow prescribes a rule and expects people to follow it. The division between healthy people and COVID-19 patients may results in prejudice. For patient 18, her social isolation pulls her to desire the Other’s desire, which expands the lack of her sense of broken. She feels different from others. Therefore, her response to the desire is de-labeling. For this reason, COVID-19 patients demand recognition. Their demand turns into the desire of the Other’s desire for a healthy image.

Social isolation, an important factor of the external environment, is a main stressor during COVID-19 and has negative effects on COVID-19 patients [29]. Patients felt anxious when they talked about the experience of isolation, as shown in the following extracts:


*“I was crushed when I was isolated for the past ten days. I couldn’t sleep. I feared that I couldn’t sleep at night, too.”*

*(Patient 7, metaphor of crumbling, breaking, and falling apart)*



*“When I started my isolation, I felt the atmosphere was something like that I had been isolated from the outside world.”*

*(Patient 10, link metaphor)*



*“I had been told that I was cured when I went back home. But a few days later I was sent back to the hospital and my family was forced to isolate. I think that is a little nonsense. Of course, it proved that the virus was very cunning. And our health systems had not prepared for this yet”*

*(Patient 14, animacy metaphor)*



*“I had to isolate in a long period where I was isolated from my families and friends even when I went back home. I could not bring people around me the danger of infection, because I was not sure when I would be retested positive.”*

*(Patient 29, carrying metaphor)*


The preceding narratives indicate the negative emotional states of COVID-19 patients during their isolation. The isolation widens and emphasizes the identity gap between COVID-19 patients and healthy people. COVID-19 patients’ negative emotions, which come from the evaluation of people around them, reveal their strong desire for de-stigmatizing.

It is noteworthy that the living environment is also one of the essential factors for the patients’ psychological health, in particular for patients from the countryside. For instance:


*“Nobody in my hometown in village didn’t know my infection yet. Because you know that villagers, villagers are so, they have many mouths. If you say that you are fine, those people will exaggerate the details. I just don’t want to make villagers know my thing.”*

*(Patient 5, image metaphor)*



*“Many people discriminated against us. They kept off when they saw us. The gossip was really…unbelievable. I felt disgusted. But I could not live in others people’ eyes. Now we live in our own house other than theirs.”*

*(Patient 16, image metaphor)*


Patients 5 and 16 were living in a rural district. They showed negative emotions when they talked about people around them. Although the village has developed rapidly in the past few years, villagers remain conservative in their treatment of COVID-19 infected persons. These people mainly focus on the negative effects of COVID-19. The fear of contagion caused their discriminatory attitude toward COVID-19 patients. This discrimination forced COVID-19 patients to seek a way to reach the desire of the Other, namely, to reach the concept in the “mirror” in which a perfect ego exists.

#### 4.2.3. The Efforts of Self-Construction for the Sense of Broken

The self is a key issue in Lacan’s theory because it is related to a philosophical problem related to how people form a concept of “self”. To distinguish “self” from “other”, people need to recognize “I”. The issue inevitably pertains to the external environment. One of the fundamental criteria for a subject’s existence is their ability to differentiate his/her Self from the surrounding environment [30]. If people wish to observe himself/herself, he/she must observe the self as an observer like other people. People can imagine themselves with the help of mirror. Lacan contends that infants constantly change and adapt to the external world in their growth [19]. Parents put the infants in a social environment in which they have to face many issues. Thus, the interrelation between “I” and “the world” is significant in the whole system. Two dual fantasies, namely of myself and of the world, are inseparable in Lacan’s model. Desire is present in both of the fantasies as the gap that separates the fantasy from the reality it seeks to capture [31].

In our cases, the COVID-19 patients’ psychological process of self-construction can be described in light of the conflict between freedom and constraint, as well as the re-labeling of the ego.

The first point concerns the conflict between freedom and constraint. The corresponding behaviors are labeling and de-labeling. COVID-19 patients used “free” to label recovery and drew on “jail” or “prison” to label isolation. For instance:


*“I was so happy when I was discharged from hospital. I couldn’t breathe the air outside because I was isolated in the hospital. I could not go outside as both doors and windows were closed. When I was discharged from the hospital, I thought that the air was clean, and humans were free.”*

*(Patient 7, image metaphor)*



*“Yes, I felt I was free from the jail. The air was clean. Words could hardly describe my happiness. I relaxed finally.”*

*(Patient 22, image metaphor)*



*“Patients were not allowed to go out because each ward was locked. We had limited space of activity. …I think it was like a prison.”*

*(Patient 3, image metaphor)*


Patients’ transitioning between constraints and freedom demonstrates the transformation between labeling and de-labeling. In the process, COVID-19 patients value their identity as a healthy person. Their behaviors point to the construction of such identity. A person’s identity is rooted in others’ characterization of that person in light of the social conventions and constructs of the cultures and traditions, which have shaped the personal identity of the ones who make the attributions [32]. Apparently, COVID-19 patients confirm their identity with respect to others’ evaluation since they are the concrete parts of the Other. Consequently, labeling and de-labeling occur. The examples below illustrate the de-labeling phenomenon:


*“That means some managers with power felt uncomfortable. I think at that time what should I do was to change their mind and to make them not identify me as a sensitive object but a normal person. And I supposed that was what I thought most. My trouble was to find the way of solution.”*

*(Patient 1, image metaphor)*



*“The doctor was nervous when the result was positive. Then I was kept in a big room. No one answered me when I called them or asked them for treatment. They put the dinner at the door and ran away soon. I felt a sense of discrimination. I was discriminated, and I was locked in a room in which nobody took some measures to deal with it.”*

*(Patient 14, behavior of others)*


COVID-19 patients are assessed as being harmful and dangerous since the virus is contagious. Such negative evaluations put patients in an inferior social position. Therefore, patients take positive actions to de-label their identity. As the example below shows, to obediently follow the prescribed treatment in the hospital is an effort in de-labeling.


*“The doctor told me how to do this, and I would follow his demands. I would follow his way for treatment. I won when the doctor pronounced his victory.”*

*(Patient 28, fighting or battling)*


While COVID-19 patients practice de-labeling, labeling is equally important. They label themselves with strong tendentious behaviors to compensate for the negative effects due to the patient identity, as shown in the following example:


*“Our country is very strong. I’m touched so much because doctors and nurses took good care of me and treated my illness carefully. They did not take me as a patient in the hospital where I felt the warmness of home”*

*(Patient 11, temperature, family)*


Furthermore, the emotion of patient fear regarding hospital quarantine is a manifestation being labelled as a normal person rather than as a patient. For example:


*“When I was transferred to another hospital, I…I didn’t know what happened. The transfer let me have no bottom in the heart. I was really afraid. I went there to be an experimental object, for real.”*

*(Patient 10, container)*


In the process of labeling and de-labeling, COVID-19 patients are looking for the change of identity. As Drummond claims, the change of self-identity may be affected by physiological change due to illness, injury, or a transitory change in mood [32]. It is the wish of changing identity that results in a strong sense of loss. For instance:


*“Because I had never experienced such things, I had a feeling of fear. I feared of losing everything, and I feared of leaving the beautiful world.”*

*(Patient 3, closeness and distance)*


The sense of loss is a symbol of lack. Reality breaks patients’ past self-constructed identity. They lose the shell (i.e., “a healthy person”) which wraps the fragile self. Subsequently, a sense of broken emerges in patients’ hearts.

The second point is related to maintaining a perfect ego and re-labeling of the ego. In our interviews, COVID-19 patients emphasized that they are healthy and harmless after recovery. In their mind, there is a “perfect ego” existing in the place where the Other is, namely, the place of lack. People around the subject are the symbol of the Other. The whole system seems like a huge mirror which reflects the perfect ego of the subject (i.e., COVID-19 patients). People and society comprise the mirror for COVID-19 patients. They use words to prescribe what a normal person should do and what a patient should do. In this regard, fear of COVID-19 may strike the system, and anxiety pushes COVID-19 patients to de-stigmatize. For COVID-19 patients, the desire emerges from the sense of lack. They want to rebuild the broken self through their behaviors of labeling and de-labeling. They attempt to recollect the broken self and fit it to the image of the perfect ego in the mirror, namely, “a healthy person” in reality. Then, patients re-label the image in the mirror (although it is a delusory image) to integrate their broken self. Consequently, COVID-19 patients constantly emphasize that they are healthy, as illustrated in the following example:


*“You know that we left and came back for a long time. We were always shouting. Our hearts were shouting that we were not patients. We did not get ill. But we were diagnosed as patients because we matched the condition of COVID-19 test and diagnosis under the system and policies.”*

*(Patient 30, animacy)*


COVID-19 patients rebuild the broken self to fit and establish a new and perfect self through denying the identity of patient. The nature of the process is to recognize the ego and to be recognized as a healthy person. In this sense, desire plays a role between “I” (recognize the ego) and the Other (be recognized by the Other).

## 5. Discussion

The present study analyzed how metaphors, emotions, and behaviors in lived experience narratives revealed the psychological states of COVID-19 patients from the perspective of Lacan’s desire theory. We analyzed how the desire drove patients to re-integrate the self to deal with psychological trauma attributable to COVID-19 infection. In the process, the Other sets the rules regarding “health” in the whole social system. The desire of lack pushes COVID-19 patients to follow the rules and to rebuild the broken self in three levels of the desire: the desire of the Other’s desire (observe the rules), the desire for the Other (the lack of desired object), and the desire to be the object of desire (being healthy, recognize and be recognized). In addition, we addressed how COVID-19 patients fulfill the concept of a “perfect ego” to satisfy the delusory desire through labeling and de-labeling. While desire itself can never be satisfied, the process of desire is somehow satisfied. In doing so, COVID-19 patients build an image of a “misplaced” healthy person eventually.

Generally, COVID-19 patients in the present study are in a sense of broken during the long journey of recovery due to their anxiety regarding their patient identity. Their desire struggles between self-recognition and recognition by others, which impels patients to change their identities. The desire underlying patient narratives and behaviors points to recognition, in that discrimination-related anxiety posed serious mental health consequences among patients. Thus, recognition is vital to help COVID-19 patients to attain the meaning of the self.

The psychological problems of COVID-19 patients in our study, including fear, anxiety, insomnia, stigma, and depression, are consistent with those of previous investigations [9,11,12,33,34,35,36,37]. It is noteworthy that discrimination from the external social environment has increased the negative emotion of COVID-19 patients. The discrimination is largely caused by the negative consequences of COVID-19. In the literature, negative use of metaphors in media narratives evokes negative effects of COVID-19 [38,39]. Unscientific information regarding the infection and transmission of COVID-19 causes fear among the public, and makes people exaggerate the negative attributes of the virus. Social discrimination against COVID-19 patients is likely to be a manifestation of public anxiety on the one hand, and on the other, the unfriendly behaviors of some people (e.g., the rejection and stigmatization of COVID-19 patients) establish an inappropriate correlation between the infected patient and the virus. This leads to severe anxiety and self-stigma among COVID-19 patients. The research findings in the present study suggests that COVID-19 patients, as the vulnerable group, should receive more attention to maintain their mental health [10,13,15,40].

To deal with the negative effects, COVID-19 patients in the present study tend to strengthen their identities with positive information. For instance, they use positive metaphors when talking about the role of the country and the medical personnel, which revealed their demands. This is consistent with observations of previous research regarding the positive function of metaphors in communication about the COVID-19 pandemic [22,23,38,39].

The complicated lived experience and psychological processes of COVID-19 patients enable them to speak and behave in a specific pattern to reflect the interaction between “I” (COVID-19 patients) and “the Other” (the environment or external world), which can be seen in the darkness and light metaphor, freedom and prison metaphor, and, on the one hand, patients’ behaviors against discrimination, and on the other, cooperation with pandemic policies. The clinical implication of this is that we can reveal the drives of COVID-19 patients’ desire of “being a healthy person”, and thus support them psychologically. We can provide suggestions for COVID-19 patients to mitigate or to restrain the strong negative effects of lack-related desire so that they can rid themselves of the shadow of the past experience. Velykodna illustrated reflections on countertransference related to the real loss during the COVID-19 pandemic [16]. For psychoanalysts, countertransference plays an important role in psychological interventions for COVID-19 patients. The mechanism of dreams is also useful in dealing with trauma during COVID-19 [41]. Specifically, the descriptive narratives of dreams demonstrate potential continuity by creating links between personal and collective experience, as the past, present, and future converge, offering therapeutic possibility. The literature reveals two potential markers in the psychological support for COVID-19 patients during the pandemic. One is the relationship between the guide (e.g., psychiatrist) and the patients. A guide has significant effects on patient emotional regulation, while the emotional narratives or behaviors of patients point to the cause–effect relation of their psychological symptoms. The other is that factors in reality, such as interpersonal relationships and community policies, should be considered in the social and psychological support of COVID-19 patients.

Based on the underlying meaning of emotions and behaviors in the psychoanalysis of COVID-19 patient narratives, the present study is insightful in providing practical suggestions to deal with the psychological problems of the COVID-19 patients from three perspectives, namely, the relationship between psychological supporters and patients, the interpersonal relationships of patients, and the living environment of patients.

The first suggestion concerns the relationship between psychological supporters and COVID-19 patients. COVID-19 patients complain that psychological support is limited because they are in a poor psychological condition. The psychological supporter plays a role as a guide rather than a participant in the recovery experience. The patients may be suspect of the measures employed by the psychiatrist and claim that the psychiatrist is an outsider. Thus, we should avoid putting patients in a position of weakness in clinical practice. Mental health nurses (MHNs) can be introduced into the process of psychological support. Emotional intelligence capabilities have been associated with the identity of mental health nurses and as being desired attributes by users of mental health services [42]. Browne and Hurley affirmed the therapeutic value of the relationship between mental health nurses and patients, given that MHNs may be more acceptable to COVID-19 patients than psychiatrists, since the focus of work of MHNs is nursing, which may make patients feel less “strange” [43].

The second suggestion is about interpersonal relationship of COVID-19 patients. In the hospital, it is difficult for patients to achieve self-recognition due to their identity as a patient. The relationship between patients and doctors is simpler than that out of the hospital. Medical personnel play a positive role in helping patients to de-label the identity of patient; COVID-19 patients tend to use positive discourse to describe medical staff in our interviews. Furthermore, the relationship between patients and doctors can be transplanted to the relationship between the patients and their family. In family relationships, patients fail to rid themselves of the constraint of their patient identity. Thus, family members and relatives should participate in the building up the relationship based on mutual understanding and love [13,14,25]. The purpose is to give the patients a leading role in their psychological recovery. Service users need to ask how patients want family members to be involved [44].

In the present study, normal interaction between COVID-19 patients and their family, friends, and neighbors is a critical factor in helping COVID-19 patients to recognize themselves. In our interviews, COVID-19 patients were usually observed to have mood swings when they were asked to answer a question about their experiences, which suggests that they still have difficult experiences embedded in their minds although they have been cured [13,25]. This makes patients more sensitive to the environment. When patients were asked about changes, some reported that changes happened on him/her (e.g., sub 28 said that he had changed the lifestyle). Their changes can be seen as the obedience to the prescribed rules by the Other from the past to the future. Psychologically, the changes are a clue to their wish to return to the initial “me”—the image of a normal person, rather than a fake self, reconstructed after the COVID-19 infection. They can foster interpersonal relationships in a stepwise manner in different ways. For instance, contacting family, friends, and colleagues through telephone calls and video chats helps to inform people around these patients of their recovery state. With the increase of acceptance, cured patients can join social activities with family, friends, and colleagues. Meanwhile, to build an open relationship is essential to maintain the position of patients themselves. Patients may make friends with those who share similar experiences of infection and quarantine. Common lived experience can provide a cathartic way to adjust their mental states [13,25].

The third suggestion pertains to the living environment of COVID-19 patients, which can assist COVID-19 patients to be recognized by others. It is urgent to eliminate discrimination concerning COVID-19 infection [25], which may largely impact patients’ views on the external world (or more specifically, the community where they are living in). Support should be provided to make the patients believe that others are participants rather than leaders. In patients’ minds, the image of a healthy person is of a person who is not treated as a special subject (i.e., a patient). Although they are in a special position in the system, their identity should be accepted in the community and the whole social environment [24,25]. COVID-19 patients require a new self-image since the anxiety from other’s pressure threatens their image of the self. For instance, some patients in our study expressed their determination for contribution, such as via blood donation for antibodies research. Similarly, some patients had positive responses to help others when they were asked whether they wanted to share their experiences [25]. This is a signal from these patients with respect to their de-labeling.

During the interviews, the communication between the COVID-19 patients and interviewers was a process in which the patients actively seek recognition since they frequently narrated their experiences by asking “is it right?”. Patients’ long-term anxiety prompted them to seek recognition from others to appease themselves. In doing so, they can maintain a healthy and harmless image. Our analysis shows that COVID-19 patients reject discrimination with apparently negative emotions [24,25]. Discrimination brings about anxiety, which is a restrictive factor in rebuilding the self of COVID-19 patients. As a result, patients experiencing fear and anxiety cannot return to the prevailing social system successfully. Difficulties in their interpersonal relationships enhance the pressure on them and some patients can develop compulsive behaviors as a consequence [22,23]. It is the fragmented identities that show negative effects on COVID-19 patients even after they return to normal life. They worry about being a trouble to others and may constantly keep a distance from others [13,25]. When people around them treat them with prejudice, patients fall into another cycle of labeling and de-labeling. In this regard, their desire emerges in the space between the “I” (COVID-19 patients) and the external world. Patient experiences are a manifestation of thinking about self-value—the value of the subject. Essentially, COVID-19 patients should take measures to prove their value and reshape the meaning of their self. For example, doing public service such as donating blood for research, working as a volunteer, or sharing their recovery experiences with newly-infected patients can help patients recognize their contribution to community and society.

Change of social environment is equally important in promoting the process of self-rebuilding. It has positive effects in helping COVID-19 patients overcome the desire of lack. Specifically, social support from the government is necessary. Implementing policies to avoid discrimination in employment can largely reduce the anxiety of their identity as COVID-19 patients [25]. Furthermore, the personal medical information of COVID-19 patients should be protected since patients may suffer from the negative consequences of private information leakage [24]. A key point to eliminating discrimination is the appropriate and effective communication of valid scientific information regarding COVID-19 [25]. Healthcare authorities can therefore contact COVID-19 patients who meet the standard of recovery to reassure the public with respect to COVID-19 patient infectivity.

### Limitations and Future Research

The present study has some limitations. First, the study is based on a small sample of COVID-19 patient narratives, and the findings may not be representative of all COVID-19 patients’ experiences. Second, we observed the patient narratives to reveal their psychological problems in terms of psychoanalytic theories such as Lacanian theory. Given that each individual’s experience is unique and complex, phenomenal qualitative psychoanalysis may not fully capture the complexities of the psychological and social factors at play in the experiences of COVID-19 patients. Third, the study did not consider the experiences of COVID-19 patients from different cultural backgrounds, ages, or socioeconomic statuses, which may impact their experiences and perceptions of the pandemic.

Future research can gather a large sample size of patients and conduct comprehensive analysis to confirm the generalizability of the research findings in the present study. In addition, natural language processing (NLP) methods, such as sentiment analysis and topic modeling, can be employed to measure patient narratives to uncover the psychological status of COVID-19 patients. Furthermore, factors such as ages, socioeconomic background, and gender can be considered to measure the different psychological states among patients during the pandemic.

## 6. Conclusions

The present study is a psychoanalytic discourse of COVID-19 patient narratives using Lacan’s desire theory. We have shown the internal and external factors related to psychological problems seen in COVID-19 patients. Regarding internal factors, patient narratives show how desire functions at the psychological level. For the external dimension, desire emerges between “I” and “the Other”. The internal functioning of “I” (the self) can be seen to affect the objections of the self in the external world. The external world, in turn, responds to such objections. COVID-19 patients seal “I” (the identity of patient) up and throw it in reality. The aim is to reach the “perfect ego”. As patient behavior reflects psychological status, COVID-19 patient responses to external factors reflect their internal desire. Based on these two elements, our findings reveal three aspects regarding COVID-19 patient psychological problems, namely, the sensitivity of COVID-19 patients to the environment, the pressure of COVID-19 patients from the environment, and the reconstruction of the self via labeling and de-labeling.

To resolve these problems, comprehensive intervention strategies should be employed to help patients adjust to the external environment and to achieve self-recognition. Furthermore, therapists should support COVID-19 patients in dealing with pressures from the environment. Lastly, social support should be provided to help COVID-19 patients take positive steps to reintegrate into the external world. Adequate support and social recognition may not only relieve patients’ desire of lack, but also largely reduce their anxiety regarding their identity as COVID-19 patients. Our findings have clinical implications in the quest to help COVID-19 patients to adapt and reshape their identity, and to live a normal life again.

## Figures and Tables

**Figure 1 medicina-59-00712-f001:**
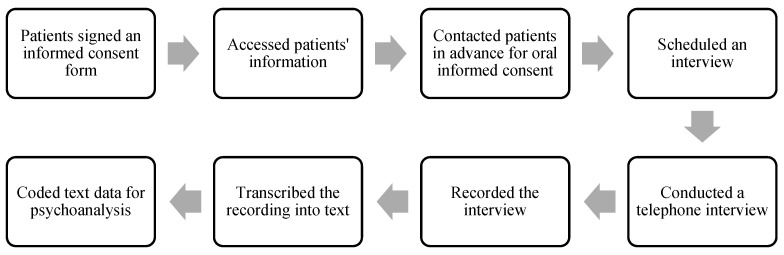
The procedure of the interview.

**Figure 2 medicina-59-00712-f002:**
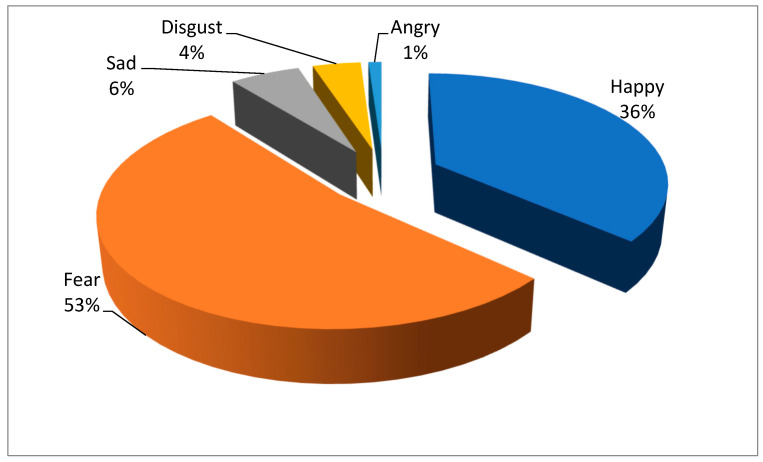
Distribution of emotional narratives.

**Figure 3 medicina-59-00712-f003:**
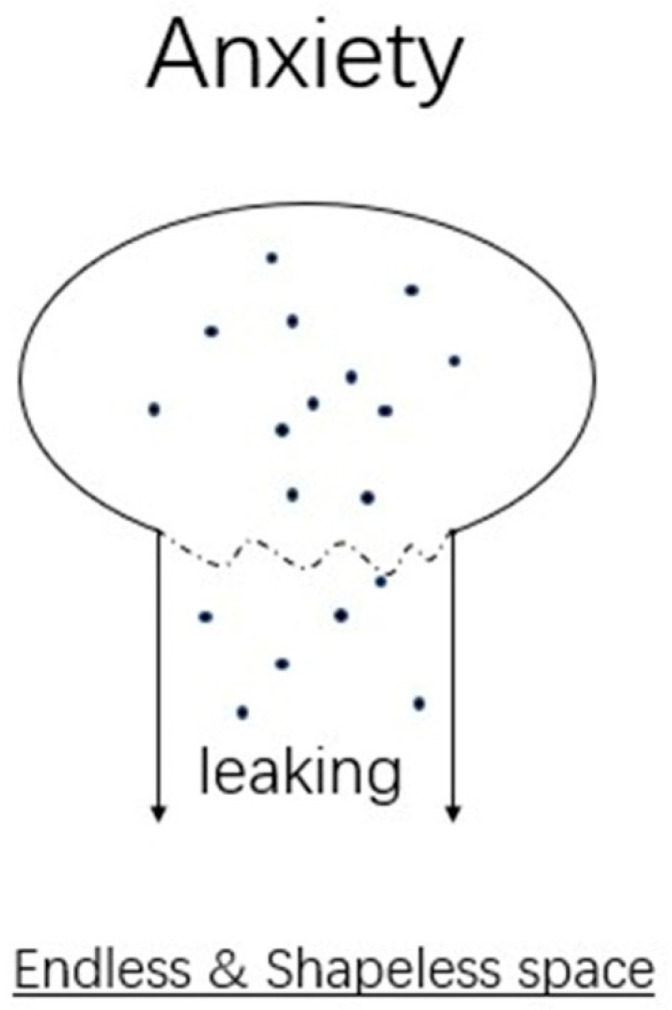
The leaking wholeness.

**Table 1 medicina-59-00712-t001:** Sociodemographic characteristics of COVID-19 patients (*n* = 36).

Characteristics	Number (%)
Gender	
Male	20 (56%)
Age	
20–30	10 (27.8%)
31–40	12 (33.3%)
41–50	8 (22.2%)
51–60	6 (16.7%)
Occupation	
Individual business	6 (16.7%)
Engineer	5 (13.9%)
Factory Worker	5 (13.9%)
Farmer	4 (11.1%)
Unemployed	4 (11.11%)
Waiter	2 (5.5%)
College Student	2 (5.5%)
Manager	1 (2.7%)
Medical staff	1 (2.7%)
E-commerce	1 (2.7%)
Housewife	1 (2.7%)
Retired	1 (2.7%)
Designer	1 (2.7%)
Finance staff	1 (2.7%)
Teacher	1 (2.7%)
Marital Status	
Married	26 (72.2%)
Unmarried	8 (22.2%)
Divorced	2 (5.6%)
Mean Hospital Quarantine Time	16 days

**Table 2 medicina-59-00712-t002:** The categories of metaphorical narratives.

Category	Instances	Percentage	Category	Instances	Percentage
Image metaphor	76	24.68%	Physical injury	3	0.97%
Motion	24	7.79%	Presence, accompanying and absence	3	0.97%
Container	16	5.19%	Journey	3	0.97%
Family	16	5.19%	Weight	3	0.97%
Life and death	15	4.87%	Depth	2	0.65%
War	14	4.55%	Liquid-based metaphors	2	0.65%
Animacy	11	3.57%	Different realities	2	0.65%
Symbolic metaphorical enactment	10	3.25%	Sense of touch	2	0.65%
Integrative behavior	10	3.25%	Size	2	0.65%
Darkness and light	9	2.92%	Spatialization	2	0.65%
Pressure	7	2.27%	Agency	1	0.32%
Carrying	7	2.27%	Balance	1	0.32%
Color	6	1.95%	Cleanliness, dirtiness	1	0.32%
Violence and impact	6	1.95%	Conduit metaphor	1	0.32%
Animal	6	1.95%	Divided self	1	0.32%
Closeness and distance	5	1.62%	Explosion	1	0.32%
Different realities	5	1.62%	Fighting or battling	1	0.32%
Temperature	5	1.62%	Finding and losing	1	0.32%
Body-related metaphor	4	1.30%	Going back and forth	1	0.32%
Crumbling, breaking, falling apart	4	1.30%	Inside and outside	1	0.32%
Fairness, justice	4	1.30%	Sense	1	0.32%
Machine	4	1.30%	Trials, law	1	0.32%
Games, chance, sport	3	0.97%	Pushing and pulling	1	0.32%
Hiding	3	0.97%	Seeing	1	0.32%

## Data Availability

The data presented in this study are available on request from the first author or corresponding author. The data are not publicly available due to privacy and ethical reasons.

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
