# Peer review of "Psychoanalysis of COVID-19 Patient Narratives: A Descriptive Study"

_medicina, 2023, doi:10.3390/medicina59040712_

Round 1

Reviewer 1 Report

1. I feel that the subject matter of Covid-19 narratives is an important one, perhaps often overlooked in medical settings and even psychiatric settings, the latter where we focus more on treating symptoms and possibly disorders.

I am not a Lacanian therapist, but rather a medical cognitive therapist (i.e., CBT plus medications) but there are common elements to all psychotherapies. Many readers will not be Lacanian therapists either, so I thought the introduction to the field you presented was a good one.

2. Line 23 -- better define what you mean as "sensitivity to the environment". (It is only the Abstract, but even so, given these are your findings it is important for the reader of the Abstract to understand what you mean exactly.)

3. Lines 41-42 -- Specify if you mean post-Covid, which is what I think you are saying. Also, post-Covid, in the most common mental health problems, issues of cognition should be included, i.e., fatigue, memory problems, difficulty concentrating.

4. Lines 130 ....   Results section -- I believe the results section has to be more in line with the scientific method, i.e, better specify what the results of your study was. It is too spread out and vague.

5. Line 439 -- Why is this section Discussion also given the heading "3" which was the number for the Results section.  I find the Results Section and the Discussion Section not to be ready for publication. Please think about the scientific method. What are you measuring for the Results section? And in the discussion section clarify the results from the previous section, i.e., discuss them.

Table 1 is good.... giving me some information. However, it is *not* good if you do nothing with this information. Do not give me information that has no relation with the results. If there is no relation with the results you can just summarize average age, etc. However, is there a relation?

Figure 2 is good. You show Coded data for psychoanalysis at the end. But why do you stop here?  Where is this data?  What is this data? What relationships does this data show?

I do not want to see hundreds of pages of transcripts from patients -- I want to see results which you have processed, i.e., you have measured something from all these narratives (if you have not measured anything then it is not science and it should not be published in this journal).

In the Abstract you give findings #1, #2, and #3. These should more clearly be shown in the text.

Again, this is a medical journal, i.e., it needs to follow the scientific method. As research therapists, information must be extracted from the narratives, and in your scientific article you should show what these measured values indicate.

Reviewer 2 Report

 1.      The main question addressed by the research is Measurement the subiectivity od covid-19 syndrome.

 2.      the topic original in the field and specific gap is connected with new look at subjectivity syndrome  [SS] based on psychoanalytic framework.

3.Compared with other published materia, probably this research in a more large form describe SS. added to the subject area

4.The specific improvements should the authors consider regarding the methodology: Application of psychoanalytical approach/method and intterpretation of the SS component

5..The conclusion are consistent with theoretical framework and the results

6.. the references are appropriate, because this is a rather new approach

 7. No specific comments on the tables and figures, because empirical data are appropiately presented in the arcicle.

Author Response

  1. The main question addressed by the research is Measurement the subiectivity of covid-19 syndrome.

Response: Yes, this study aims to explore the subjective syndrome of COVID-19.

  1. The topic original in the field and specific gap is connected with new look at subjectivity syndrome  [SS] based on psychoanalytic framework.

Response: Yes, we try to use Lancan’s desire theory to present COVID-patient subjective syndrome to reveal their psychological problems.

  1. Compared with other published material, probably this research in a more large form describe SS. added to the subject area

Response: Thanks for the positive comments. Covid-19 narratives are an important marker for mental health of patients. This subjective side of patients is perhaps often overlooked in medical settings and even psychiatric settings, the latter focus more on treating symptoms and possibly disorders. We hope this search can added to the subject area compared to other published literature, as noted by the reviewer.

  1. The specific improvements should the authors consider regarding the methodology:Application of psychoanalytical approach/method and interpretation of the SS component

Response: In introduction section, we have added information on the methodology of the study regarding the Application of psychoanalytical approach/method and interpretation of the SS component explanation, as copied below:

“This study employed semi-structured telephone interviews with 36 individuals infected with COVID-19 in China by convenience sampling from December 2020 to March 2021 to collect lived experience narratives. The phenomenological qualitative interview method can offer insights into patients’ subjective syndromes [13, 14]. Emotions, metaphors, and behaviors in interview data were coded for psychoanalysis in line with Lacan’s desire theory, in order to reveal patients’ psychological problems.”

In section 3.1, we have added a sentence: “The psychoanalysis of patient narratives can reveal subjectivity syndrome of COVID-19 infected persons.”

In Section 3.4, we also mentioned the SS in the methodology: “These narratives can demonstrate subjectivity syndrome of patients due to COVID-19 infection.”

  1. The conclusions are consistent with theoretical framework and the results

Response: Thanks for the positive comments

  1. The references are appropriate, because this is a rather new approach

Response: Thanks for the positive comments

  1. No specific comments on the tables and figures, because empirical data are appropriately presented in the article.

Response: Thanks for the positive comments

Reviewer 3 Report

Dear Authors,

the paper presents a psychoanalytic investigation of COVID-19 patients, utilizing Lacan's desire theory to explore the way patients' desires are presented in their lived experience narratives. The study included 36 COVID-19 patients in China who were interviewed about their experiences of COVID-19 infection, and emotions, metaphors, and behaviors in patient narratives were collated as the main points for psychoanalysis.

The findings of the study highlight the sensitivity of COVID-19 patients to their environment, their desire to be healthy, and the pressure they face from the environment. The study also revealed patients' responses to the external world, both positive and negative, and how they construct their own image of a healthy person. The study concludes that COVID-19 patients have a psychological need to get rid of the identity of a "patient" at the individual and social level and suggests clinical implications to help them reshape their identity and live a normal life.

Overall, the paper provides an interesting and important perspective on the psychological aspects of COVID-19 patients' experiences, and the use of psychoanalytic methods to investigate these experiences is unique and valuable. However, the paper has major issues that must be addressed. 

1.

While the introduction provides a good overview of the literature and the study's purpose, there are some academic issues that should be addressed:

The introduction could benefit from a clearer and more explicit statement of the research question and objectives. The current introduction provides a good overview of the background and context, but it is not entirely clear what the specific research question is or what the study aims to achieve.

The introduction could be strengthened by a clearer explanation of the theoretical framework and how it relates to the study. While the section on Lacanian Desire Theory is well explained, it could be more explicitly linked to the study's focus and objectives.

The introduction primarily relies on previous research to justify the need for the current study, but it would benefit from a more detailed explanation of the gaps in the literature that the study aims to fill. Specifically, it could be more explicit about what is missing from the current literature that the study aims to address.

There is a lack of information on the sample population used in the study. While it is mentioned that the study included 36 COVID-19 patients in China, there is no information on how the patients were recruited, the criteria for inclusion/exclusion, or any demographic information.

The introduction does not provide any information on the methodology of the study, including the design, data collection, or data analysis. This lack of information makes it difficult to assess the validity and reliability of the study's findings.

Overall, the introduction could benefit from more explicit statements of the research question and objectives, a clearer explanation of the theoretical framework, and more detailed information on the methodology and sample population.

Section 2

The section presented appears to be an excerpt from a research paper on the impact of the COVID-19 pandemic on patients' mental health. While the author draws on Lacan's concept of ego and desire to interpret COVID-19 patient narratives, several weaknesses and issues can be identified in the text:

Lack of clarity: The language used in the section is often overly complex and abstract, which makes it difficult to follow the author's argument. The use of technical terms from psychoanalysis and philosophy may make it hard for readers who are not familiar with these fields to understand the text.

Lack of empirical evidence: While the author draws on COVID-19 patient narratives to illustrate their argument, there is a lack of empirical evidence to support the claims being made. The section does not provide any statistical data or research findings to back up the author's interpretation of the patient narratives.

Limited generalizability: The section is based on a small sample of COVID-19 patients, and the findings may not be generalizable to other populations. The study does not take into account the experiences of COVID-19 patients from different cultural backgrounds, ages, or socio-economic statuses, which may impact their experiences and perceptions of the pandemic.

Lack of ethical considerations: The study does not address ethical considerations in collecting and analyzing patient narratives. For instance, it is not clear how the patient narratives were obtained, whether the patients were informed about the research purpose, and whether they provided informed consent. The study also does not address issues of confidentiality and anonymity in reporting patient narratives.

Oversimplification of complex issues: The section oversimplifies complex issues related to mental health and the COVID-19 pandemic. For instance, the section implies that patients' desire to be recognized by others is solely related to the external environment and the desire of the Other, without taking into account the internal factors that may contribute to patients' psychological symptoms. The text also suggests that the patients' shame and anxiety are solely related to their identity as COVID-19 patients, without acknowledging the impact of other social and economic factors.

Overall, the section could benefit from a clearer and more concise presentation of the research findings and a more nuanced interpretation of the patient narratives. The study could also be strengthened by addressing ethical considerations and limitations in the research design.

section 3

he section provides a clear description of the research design, participants, procedure of interviews, and data analysis. However, it is important to consider the study's limitations and potential biases, which may affect the generalizability and validity of the findings.

All the best

Round 2

Reviewer 1 Report

line 92 – “due to the desire for lack”  -- does not make sense – please better explain for the reader who is not a Lacan therapist. This phrase occurs again, i.e., line 655.

line 292 – Figure 3 quality is poor – should be improved, i.e., higher resolution.

Author Response

line 92 – “due to the desire for lack”  -- does not make sense – please better explain for the reader who is not a Lacan therapist. This phrase occurs again, i.e., line 655.

Response: Sorry for the confusing phrase. We have explained “desire for lack” with a bracket in line 92, as copied below:

“...It is hypothesized that COVID-19 patients display abnormal behaviors and mental health symptoms due to the desire for lack (i.e. desire for a fulfilling of the lack, or desire for that which they lack)

line 292 – Figure 3 quality is poor – should be improved, i.e., higher resolution.

Response: We have revised Figure 3 and enhanced the resolution.

Thank you very much.

Reviewer 3 Report

I would like to thank the authors for addressing the major issues within the paper. However, There are a few issues. One potential weakness at line 293 is that the author does not fully explain the connection between patients' behaviors and their desire for recognition. While the author suggests that patients engage in repetitive behaviors to attain the perfect ego and be recognized by others, they do not fully explore how these behaviors achieve that goal. Additionally, the author relies heavily on patient narratives to support their argument, which may not be representative of all COVID-19 patients' experiences. Further research and analysis would be needed to confirm the generalizability of these findings.

One potential weakness in this section also is that it may oversimplify the experiences and emotions of COVID-19 patients. While it is true that many patients may feel shame and anxiety due to the stigma associated with the virus, and that external factors such as fear of contagion can exacerbate these negative emotions, it is important to acknowledge that each individual's experience is unique and complex. Additionally, the passage may rely too heavily on psychoanalytic theories such as Lacanian theory, which may not fully capture the complexities of the psychological and social factors at play in the experiences of COVID-19 patients.

Author Response

I would like to thank the authors for addressing the major issues within the paper. However, there are a few issues.

One potential weakness at line 293 is that the author does not fully explain the connection between patients' behaviors and their desire for recognition. While the author suggests that patients engage in repetitive behaviors to attain the perfect ego and be recognized by others, they do not fully explore how these behaviors achieve that goal.

Response: Thanks for the comments. We have added a paragraph to explain patients' behaviors and their desire for recognition, as copied below:

"…When patients’ desire for recognition (“a healthy person” in reality) could not be fulfilled in the external world, they were likely to seal “I” (the identity of patient) up in the internal world. Their anxiety of lack may then turn into a series of repetitive behaviors in the outside world, which reflected patients’ internal desire for a fulfilling of the lack (i.e. “perfect ego”)."

Additionally, the author relies heavily on patient narratives to support their argument, which may not be representative of all COVID-19 patients' experiences. Further research and analysis would be needed to confirm the generalizability of these findings.One potential weakness in this section also is that it may oversimplify the experiences and emotions of COVID-19 patients. While it is true that many patients may feel shame and anxiety due to the stigma associated with the virus, and that external factors such as fear of contagion can exacerbate these negative emotions, it is important to acknowledge that each individual's experience is unique and complex. Additionally, the passage may rely too heavily on psychoanalytic theories such as Lacanian theory, which may not fully capture the complexities of the psychological and social factors at play in the experiences of COVID-19 patients.

Response: Thank you for raising these issues. We acknowledge these limitations in the “Limitations and Future Research” section according to the reviewer’s comments. The updated limitations and future research directions are copied as below:

Limitations and Future Research

    The present study has some limitations. First, the study is based on a small sample of COVID-19 patient narratives, and the findings may not be representative of all COVID-19 patients' experiences. Second, we observed the patient narratives to reveal their psychological problems in terms of psychoanalytic theories such as Lacanian theory. Given that each individual's experience is unique and complex, the phenomenal qualitative psychoanalysis may not fully capture the complexities of the psychological and social factors at play in the experiences of COVID-19 patients. Third, the study did not consider the experiences of COVID-19 patients from different cultural backgrounds, ages, or socio-economic statuses, which may impact their experiences and perceptions of the pandemic.

    Future research can gather large sample size of patients and conduct comprehensive analysis to confirm the generalizability of the research findings in the present study. In addition, natural language processing (NLP) method, such as sentiment analysis and topic modeling, can be employed to measure patient narratives to uncover the psychological status of COVID-19 patients. Furthermore, factors such as ages, socio-economic background, and gender can be considered to measure the different psychological states among patients during the pandemic.

Thank you very much.